

# The TropD software package: Standardized methods for calculating Tropical Width Diagnostics

Ori Adam[1], Kevin M. Grise[2], Paul Staten[3], Isla R. Simpson[4], Sean M. Davis[5,6], Nicholas A. Davis[5,6], Darryn W. Waugh[7], and Thomas Birner[8,9]

[1]Hebrew University of Jerusalem, Israel
[2]University of Virginia, USA
[3]Indiana University, USA
[4]National Center for Atmospheric Research, USA
[5]NOAA Earth System Research Laboratory Chemical Sciences Division, USA
[6]Cooperative Institute for Research in Environmental Sciences, University of Colorado, USA
[7]Department of Earth and Planetary Sciences, Johns Hopkins University, Baltimore MD, USA
[8]Colorado State University, USA
[9]Meteorologisches Institut, Ludwig-Maximilians-Universität, Munich, Germany

**Correspondence:** Ori Adam (ori.adam@mail.huji.ac.il)

**Abstract.** Observational and modeling studies suggest that Earth's tropical belt has widened over the late 20th century and will continue to widen throughout the 21st century. Yet estimates of tropical width variations differ significantly across studies. This uncertainty, to an unknown degree, is partly due to the large variety of methods used in studies of the tropical width. Here, methods for eight commonly-used metrics of the tropical width are implemented in a Tropical-width Diagnostics code

package (TropD) in the MATLAB programming language. To consolidate the various methods, the operations used in each of the implemented methods are reduced to two basic calculations: finding the latitude of a zero crossing, and finding the latitude of a maximum. A detailed description of the methods implemented in the code and of the code syntax is provided, followed by a method sensitivity analysis for each of the metrics. The analysis provides information on how to reduce the methodological component of the uncertainty associated with fundamental aspects of the calculations, such as monthly vs. seasonal averaging

biases, grid dependence, sensitivity to noise, and sensitivity to threshold criteria.

## 1  Introduction

Theoretical and climate-modeling studies suggest that the tropics widen in response to global warming (e.g., Lu et al., 2007; Levine and Schneider, 2015; D'Agostino et al., 2017). Yet estimates of the observed widening rates in recent decades are highly

uncertain – between 0 to 2 degrees latitude per decade (e.g., Davis and Rosenlof, 2012). A considerable part of this uncertainty is due to a profusion of methodologies for calculating the width of the tropics, which obfuscates the actual variations across observational and modeling datasets. The goal of this work is to help reduce the methodological component of the uncertainty



in studies of tropical width variations by providing standardized calculation methodologies, optimized for the present climate, for commonly used diagnostics.

The standardized methodologies are implemented in a Tropical-width Diagnostics code package (TropD, doi:10.5281/zenodo.1157043) in the MATLAB programming language (The MathWorks), which can be used generically across datasets. We present methodologies for each of the following categories of tropical width metrics:

1. $PSI$ – the subtropical edge of the tropical circulation delineated by the meridional mass streamfunction

2. $TPB$ – the latitude of the subtropical tropopause break

3. $OLR$ – the subtropical latitude where outgoing longwave radiation crosses a certain threshold

4. $STJ$ – the latitude of the subtropical jet

5. $EDJ$ – the latitude of the midlatitude eddy-driven jet

6. $PE$ – the subtropical latitude where precipitation-minus-evaporation becomes positive

7. $UAS$ – the subtropical latitude where the zonal-mean near-surface wind becomes westerly

8. $PSL$ – the latitude of the subtropical sea-level pressure maximum

We show that the operations required for all of the methodologies in all of the metric categories listed above can be reduced to two basic calculations:

(i) Calculating the latitude of the zero-crossing of a given field

(ii) Calculating the latitude of the maximum of a given field.

In section 2 we provide technical guidelines for these two basic calculations and provide general information on TropD. In section 3 we provide technical guidelines for each of the eight metric categories listed above. In section 4 we analyze the sensitivity of the metrics to the choice of methodology using monthly zonal-mean data derived from the European Center for Medium-Range Weather Forecasts (ECMWF) Interim Reanalysis (hereafter ERAI; Dee et al., 2011) and from historical simulations of 34 models participating in the fifth phase of the Coupled Model Intercomparison Project (CMIP5, Table 1). We conclude in section 5.

## 2 Basic calculations

### 2.1 Data and code structure

Although some of the metrics presented here may be used on zonally varying fields, we stress that the methodologies described here are designed for use on zonal mean fields (the code has not been tested on zonally varying fields). Calculations in the





TropD software assume pressure-latitude (hPa, latitude degrees) coordinates where the pressure level closest to the top of the atmosphere and the latitude gridpoint nearest to the southern pole are the first elements in the vertical and meridional ordinates, respectively. To reduce sensitivity to format variations across datasets, this ordering is automatically enforced in TropD.

TropD is divided into auxiliary calculation functions, generically named `TropD_Calculate_[FunctionName]`, and
metric functions, named `TropD_Metric_[MetricName]`. Example code is provided in the file `TropD_Example_Calculations`. TropD includes monthly zonal-mean data and pre-calculated metrics derived from ERAI (for default values of the metric functions) which can be used to run the example code and validate calculations on different machines or versions of the programming language.

## 2.2 Calculating the latitude of zero crossing

The calculation of the zero-crossing latitude of some function can be generalized to the crossing of any cutoff value by raising or lowering the function by a constant. Therefore, all calculations involving cutoff criteria are translated in TropD to the basic operation of calculating the zero-crossing latitude of some field.

The following guidelines are implemented in calculations of the latitude of zero crossing:

(i) Unless the zero crossing occurs at a gridpoint, the exact latitude of the zero crossing is calculated using linear interpola-
tion between the two nearest data points on either side of the zero crossing.

(ii) In cases where multiple zero-crossing latitudes exist, the first zero crossing along the input interval is chosen.

(iii) In cases where multiple zero-crossing latitudes exist, the calculation can be defined as invalid if the latitudinal spacing between the first zero crossing along the input interval and the second zero crossing of the same sign change is smaller than some defined value.

**Comments on the code**

The zero-crossing latitude is calculated in TropD using the syntax:

» `ZC = TropD_Calculate_ZeroCrossing(F,lat,Lat_Uncertainty)`

where `ZC` denotes the first latitude of zero crossing (i.e., sign change) of the field `F` along the interval `lat` as illustrated in Fig. 1. The input parameter `Lat_Uncertainty` is intended for cases where multiple zero-crossings exist (optional and equal to
25 zero by default). It specifies the minimal allowed distance between the first and second zero-crossing latitudes of the same sign change ($\Delta y$ in Fig. 1) along the interval `lat`. In the example shown in Fig. 1, for `Lat_Uncertainty=10` (°), `ZC` is output as "Not a Number", NaN, if $\Delta y < 10$, and as the first zero crossing along `lat` if $\Delta y \geq 10$. Likewise, `ZC` is output as NaN when a zero crossing does not exist along the interval `lat`.





## 2.3 Calculating the latitude of the maximum

To account for potential noise in the data and to reduce grid dependence, the latitude of the maximum $\phi_{max}$ of some field $F$ is calculated using

$$\phi_{max} = \int_{\phi_1}^{\phi_2} F(\phi)^n \phi \mathrm{d}\phi \Big/ \int_{\phi_1}^{\phi_2} F(\phi)^n \mathrm{d}\phi \qquad (1)$$

where $\phi$ denotes latitude, $\phi_1$ and $\phi_2$ denote meridional boundaries, $F$ is positive everywhere, and $n \geq 1$ (Adam et al., 2016). For $n = 1$, Eq. (1) yields the centroid of $F$ (e.g., as in the mass-weighted wind calculation of Archer and Caldeira, 2008), and for $n \to \inf$ it yields the exact latitude of maximum of $F$. The exponent $n$ therefore acts as a smoothing parameter with maximal smoothing for $n = 1$, and no smoothing for $n \to \inf$. Based on Monte-Carlo simulations of randomized skewed Gaussian functions on randomized grid spacing and with randomized noise (an example of such a random function is shown in Fig. 2), we find that the latitude of the maximum is identified most reliably for $n \geq 6$. The dependence of the error distribution on $n$ is shown for a representative sample of 100 random functions in Fig. 3. The standard deviation of the error decreases with increasing $n$ and remains minimal for $n \geq 6$. However, the probability of large error (i.e., the probability of outlier results) increases with $n$ for $n \geq 6$.

### Comments on the code

The latitude of the maximum is calculated in TropD using the syntax:

```
» Ymax = TropD_Calculate_MaxLat(F,lat,n)
```

where `Ymax` denotes the calculated latitude of the maximum, and `F` is some field along the interval `lat` as illustrated in Fig. 2. In order to avoid rounding errors and in order to make the field `F` positive everywhere, `F` is normalized between zero and one prior to applying Eq. (1), which is calculated using trapezoidal integration along `lat`. The input field `F` is therefore not required to be positive everywhere. In addition, `F` may include NaN values; i.e., TropD ignores NaN values in the integral of `F` in Eq. (1). The exponent `n` is an optional input parameter ($n \geq 1$), set to 6 by default. In the various implementations of the metric methods described below, the function `TropD_Calculate_MaxLat` is employed in two possible configurations:

   (i) `max`: corresponding to `n=6` (moderate smoothing), and

   (ii) `peak`: corresponding to `n=30` (weak smoothing), which yields a latitude nearly equal to the latitude of absolute maximum of `F` (Fig. 2).

The differences between these two configurations and the sensitivity of the different metrics to the value of `n` are discussed further in section 4.



## 3 Tropical width metrics

In this section we provide technical guidelines for common methodologies in each of the eight metric categories. We briefly introduce each of the tropical width metric categories below. For extended reviews of the physical rationale and inter relations of these metrics in various datasets see Davis and Rosenlof (2012), Solomon et al. (2016), Davis and Birner (2017), and Waugh

et al. (2018).

### 3.1   PSI – Meridional mass streamfunction

The tropical mean meridional overturning circulation (i.e., the Hadley cells) can be defined as the tropical circulation enclosed within the zero streamlines of the zonal-mean meridional mass streamfunction $\psi$. A common tropical width metric is therefore the subtropical latitude in each hemisphere where $\psi$ changes sign poleward of the tropical streamfunction extrema.

The meridional mass streamfunction satisfies the continuity equation such that

$$v = \frac{g}{2\pi a cos\phi} \frac{\partial \psi}{\partial p} \quad ; \quad \omega = -\frac{g}{2\pi a^2 cos\phi} \frac{\partial \psi}{\partial \phi}. \tag{2}$$

Here $v$ and $w$ denote the meridional and vertical (pressure velocity) components of the zonal-mean wind, $g$ is the gravitational constant, $a$ denotes Earth's radius and $p$ denotes pressure. Since the vertical velocity is not a well observed quantity, $\psi$ is commonly calculated as the vertical integral of the meridional component of the zonal-mean wind,

$$\psi = \frac{2\pi a cos\phi}{g} \int\limits_{0}^{p} v \mathrm{d}p. \tag{3}$$

For most $\psi$-based metrics, spurious uncertainty related to the representation of subsurface data in the dataset (i.e., where $p$ is larger than the surface pressure) can be avoided by ensuring Eq. (3) is numerically integrated from the top of the atmosphere. The units of $\psi$ calculated using Eq. (3) are kg s$^{-1}$ (the annual-mean intensity of the Hadley circulation is roughly $10^{11}$ kg s$^{-1}$). Divided by the density of water (1000 kg m$^{-3}$), $\psi$ is often presented in Sverdrup units ($1Sv = 10^6$ m$^3$ s$^{-1}$), which are

equivalent to $10^9$ kg s$^{-1}$.

### Methods

The most widely used $\psi$-based metric of the tropical width is the zero-crossing latitude of the streamfunction at the 500 hPa level, poleward of the streamfunction extremum in each hemisphere. In order to reduce sensitivity to vertical variations in the streamfunction, some studies vertically average the streamfunction in the troposphere (e.g., between the 400 and 600

25     hPa levels, Hu and Fu, 2007) or, assuming stratospheric contributions can be neglected, vertically average across the entire atmospheric column (e.g., Davis and Birner, 2017). Similarly, in order to avoid ambiguity due to multiple subtropical zero-crossing latitudes, the edge of the Hadley cell is defined in some studies as the first latitude at which the streamfunction decreases to some fraction (e.g., 10%) of its extremal value in each hemisphere, or to some minimal threshold value (e.g., 25Sv; Levine and Schneider, 2011).





**Comments on the code**

The streamfunction is calculated in TropD using the syntax:

» ` Psi = TropD_Calculate_StreamFunction(V,lat,lev) `

where `Psi` is the zonal-mean streamfunction, `V` is the zonal-mean meridional wind, and `lat` and `lev` are the latitude and

pressure-level vectors. The streamfunction is calculated using Eq. (3) by trapezoidal integration from the smallest to highest
pressure levels.

The PSI metric is calculated in TropD using the syntax:

» ` [Ys Yn] = TropD_Metric_PSI(V,lat,lev,method,Lat_Uncertainty,Levels) `

where `V(lat,lev)` is the zonal mean meridional wind. As in all of the metrics described below, `Ys` and `Yn` are the tropical

edge latitudes in the southern and northern hemispheres (SH and NH), respectively, and `lat` and `lev` are the meridional and
vertical ordinates, respectively. The input variable `Levels` (optional) is a scalar or two-element vector which specifies upper
and lower pressure-level boundaries (in hPa). The default value of the input parameter `Lat_Uncertainty` (optional), used
by the function `TropD_Calculate_ZeroCrossing` as described above, is zero. The PSI metric can be calculated using
several implemented methods, specified by the `method` string (optional, not required for default methods), which are:

(i)  `Psi_500` (default): the zero crossing of $\psi$ at the 500 hPa pressure level.

(ii) `Psi_500_10Perc`: the first latitude at which $\psi$ at the 500 hPa pressure level decreases to 10% of its extremal tropical
value in each hemisphere.

(iii) `Psi_Levels`: the zero crossing of $\psi$ integrated between two pressure levels, specified by `Levels`. The default values
of the lower and upper pressure levels are 700 and 300 hPa. If a single pressure level is specified by `Levels`, the metric

function will output the zero crossing of $\psi$ at the specified pressure level. For example,

» ` [Ys Yn] = TropD_Metric_PSI(V,lat,lev, 'Psi_Levels',0,[400 600]) `

will output the zero crossing of $\psi$ integrated between the 400 and 600 hPa levels. The calculation is not sensitive to the
ordering of the pressure levels in `Levels` (i.e., setting `Levels = [400 600]` or
`Levels = [600 400]` produces the same result). Similarly, setting `Levels = [500 500]` or `Levels=500`

will produce a result identical to selecting the method `Psi_500`. If the pressure levels specified by `Levels` are not a
subset of `lev`, the pressure levels closest to the ones specified in `Levels` are used in the calculation.

(iv) `Psi_500_Int`: the zero crossing of $\psi$ integrated between the top of the atmosphere and the 500 hPa pressure level.

(v) `Psi_Int`: the zero crossing of $\psi$ integrated between the top and surface.

For all of the above methods, the edge latitude is calculated as the most equatorward latitude where the method criteria is met,

poleward of the streamfunction extremum in each hemisphere.



## 3.2 TPB – Tropopause break

The tendency of the tropopause height to abruptly drop near the subtropical jet (e.g., Fig. 4a) is often used to diagnose the tropical width. The commonly accepted definition of the tropopause follows the World Meteorological Organization (WMO, 1957): The lowest point at which the lapse rate decreases to 2K per km, and remains lower than 2K per km between this level

and all higher levels within 2km. The manner in which the latter part of the WMO definition is implemented has been shown to potentially influence the evaluation of observed trends (Birner, 2010). In addition, indirect measurements of the tropopause break derived from changes in column ozone concentrations have been shown to detect secular trends consistent with thermodynamic TPB metrics (Hudson et al., 2006). However, metrics based on ozone concentrations exhibit strong sensitivity to the methodology applied (Davis et al., 2018) and are therefore not considered here.

**Methods**

Various tropopause-based methods for calculating the zonal-mean width of the tropics are found in the literature. These generally include:

   (i) The latitude of the largest negative poleward gradient in the tropopause height (e.g., Davis and Rosenlof, 2012; Davis and Birner, 2017)

(ii) The most poleward latitude where the number of days per year with tropopause heights above a certain altitude exceeds some threshold (e.g., Seidel and Randel, 2007)

   (iii) The latitude at which the tropopause height drops below a certain fixed threshold, or a threshold that depends on the mean properties of the tropical tropopause (Birner, 2010; Davis and Rosenlof, 2012)

   (iv) The latitude of maximal difference between the potential temperature at the tropopause and at the surface (Fig. 4b; Davis
and Birner, 2013, 2017)

   Each of these methodologies present potential weaknesses. For example, threshold-based metrics are sensitive to the choice of threshold values (e.g., Birner, 2010) and the latitude of maximal gradient is sensitive to noise and grid spacing (e.g., Davis and Rosenlof, 2012). It is therefore particularly important to consider the TPB metric method most suited to the data being analyzed and the physical question being addressed.

**Comments on the code**

The tropopause height is calculated in TropD using the syntax:

```
» Pt = TropD_Calculate_TropopauseHeight(T,p)
```
where `Pt` is the zonal-mean tropopause pressure (hPa) derived from the zonal-mean temperature `T(lat,lev)` and the vertical pressure levels `p(lev)` using the method described in Reichler et al. (2003). The implementation of the 2km condition in





accordance with the WMO definition is as described in Birner (2010). It is possible to output the value of some field at the tropopause level, using the syntax

» `[Pt Ht] = TropD_Calculate_TropopauseHeight(T,p,Z)`

where `Ht` is the value of the field `Z(lat,lev)`, with identical dimensions to `T(lat,lev)`, evaluated at the tropopause

pressure level (by linear interpolation).

The TPB metric is calculated in TropD using the syntax:

» `[Ys Yn] = TropD_Metric_TPB(T,lat,lev,method,Z,Cutoff)`

The above-mentioned methodologies for calculating the TPB metric can be realized in TropD by specifying the `method` string:

(i) `max_gradient` (default): the latitude of maximal poleward gradient of the tropopause pressure, using the syntax

» `[Ys Yn] = TropD_Metric_TPB(T,lat,lev,'max_gradient')`

with the smoothing parameter `n=6`.

(ii) `max_potemp`: the latitude of maximal difference between the potential temperature at the tropopause and the minimal value of the potential temperature in each latitude column (assumed to be located at the surface), using the syntax

» `[Ys Yn] = TropD_Metric_TPB(T,lat,lev,'max_potemp')`

with the smoothing parameter `n=30`.

(iii) `cutoff`: the most equatorward latitude where some field `Z`, evaluated at the tropopause level, crosses some cutoff value `Cutoff`, using the syntax

» `[Ys Yn] = TropD_Metric_TPB(T,lat,lev,'cutoff',Z,Cutoff)`

The default value of the cutoff parameter `Cutoff` is 15,000, assuming the input field `Z` is geopotential height in units of m.

The default smoothing parameter values in the `max_gradient` and `max_potemp` methods are based on the analysis described in section 4.3. For these methods, the value of n can be set as an input scalar ($n \geq 1$) after the method string; i.e.,

» `[Ys Yn] = TropD_Metric_TPB(T,lat,lev,'max_gradient',n)`

or

» `[Ys Yn] = TropD_Metric_TPB(T,lat,lev,'max_potemp',n)`.

The tropopause break latitude is calculated equatorward of 60° for all of the methods. The method described in Seidel and Randel (2007) and similar methods which require some statistical analysis of the tropopause timeseries are not explicitly implemented in TropD. Instead, the various methods implemented in TropD (e.g., the `cutoff` method) are designed to facilitate

such calculations in a manner that is consistent across analyses.




### 3.3 OLR – Outgoing longwave radiation

Due to variations in atmospheric absorption and surface temperature, the longwave radiation emitted to space maximizes in the subtropics ($\sim$270 W m$^{-2}$ in the zonal mean; Fig. 5a), coinciding with the dry subsiding branches of the Hadley circulation. This, together with the existence of direct satellite observations of OLR, has motivated the use of OLR-based metrics for

evaluating tropical width variations (e.g., Hu and Fu, 2007).

**Methods**

Common OLR-based methods for calculating the tropical width are:

(i) The most poleward latitude at which the zonal-mean OLR is equal to 250 W m$^{-2}$ (e.g., Hu and Fu, 2007; Johanson and Fu, 2009).

(ii) The first latitude poleward of the subtropical OLR maximum at which the zonal-mean OLR drops to 20 W m$^{-2}$ below its peak value in each hemisphere (Davis and Rosenlof, 2012).

**Comments on the code**

For generality, several OLR metric methods are implemented in TropD, using the syntax:

    » `[Ys Yn] = TropD_Metric_OLR(OLR,lat,method,options)`

where `OLR(lat)` is the zonal-mean OLR. The methods are:

(i) `250W` (default): the most equatorward latitude at which OLR drops below 250 W m$^{-2}$, poleward of the subtropical OLR maximum in each hemisphere.

(ii) `cutoff`: the most equatorward latitude, poleward of the subtropical OLR maximum in each hemisphere, at which OLR drops below a certain cutoff value specified by the parameter `Cutoff`:

» `[Ys Yn] = TropD_Metric_OLR(OLR,lat,'cutoff',Cutoff)`

    The default value of the parameter `Cutoff` is 250 (W m$^{-2}$; i.e., the `cutoff` and `250W` methods are identical if `Cutoff` is not specified).

(iii) `20W`: the most equatorward latitude, poleward of the subtropical OLR maximum in each hemisphere, at which OLR drops to 20 W m$^{-2}$ below the OLR maximum.

(iv) `10Perc`: the most equatorward latitude, poleward of the subtropical OLR maximum in each hemisphere, at which OLR drops to 90% of the OLR maximum.

(v) `max`: the latitude of maximal OLR in each hemisphere, calculated using `TropD_Calculate_MaxLat` with the smoothing parameter n=6. The value of n can be set as an input scalar (n$\geq$ 1) after the method string; i.e.,

    » `[Ys Yn] = TropD_Metric_OLR(OLR,lat,'max',n)`





(vi) `peak`: the latitude of maximal OLR in each hemisphere, calculated using `TropD_Calculate_MaxLat` with the smoothing parameter `n=30`.

The flexibility in the input parameters `n` and `Cutoff` in the OLR metric function is designed to enable sensitivity testing of this metric, as well as other metrics based on one-dimensional zonal-mean fields.

## 3.4 STJ – Subtropical Jet

In idealized theory, the subtropical jets form at the edges of the poleward-moving upper tropospheric branches of the Hadley circulation in each hemisphere (e.g., Schneider, 2006). This motivates the use of the latitude of the subtropical jet as an indicator of the tropical width. However, upper level winds are also strongly affected by midlatitude macroturbulence and stratospheric processes, obliging caution in associating the latitude of maximal zonal wind with the above conceptual picture of the latitude of the subtropical jet. Indeed, recent studies find that STJ-based metrics of the tropical width are weakly correlated with lower-troposphere metrics (Solomon et al., 2016; Davis and Birner, 2017). Common STJ-based metrics take into consideration the characteristics of the upper-level zonal winds in various ways, as described below.

**Methods**

Accounting for the fact that the STJs exhibit significant variations in longitude and altitude, the latitude of the STJ as an indicator of the tropical width has been generally calculated in the literature as:

(i) The centroid of the upper-level zonal wind within a specified meridional band (e.g., the vertical average of the zonal wind between the 100 and 400 hPa levels in the 15-70° latitude band, Archer and Caldeira, 2008).

(ii) The latitude of the maximum of the upper-level zonal wind (e.g., averaged between the 100 and 400 hPa levels; Davis and Rosenlof, 2012; Solomon et al., 2016).

(iii) The latitude of the maximum of the upper-level minus lower-level zonal wind. As shown in Fig. 6, the subtraction of the lower-level wind differentiates the signal of the STJ from that of the midlatitude eddy-driven jet, which is characterized by stronger vertical homogeneity (Davis and Birner, 2016, 2017).

**Comments on the code**

The STJ metric is calculated in TropD using the syntax:

```
» [Ys Yn] = TropD_Metric_STJ(U,lat,lev,method,n)
```

where `U(lat,lev)` denotes the zonal mean zonal wind. The available methods are:

(i) `adjusted_peak` (default): The latitude of the maximum of the zonal wind averaged between the 100 and 400 hPa levels minus the zonal wind at the 850 hPa level (smoothing parameter `n=30`).

(ii) `adjusted_max`: The latitude of the maximum of the zonal wind averaged between the 100 and 400 hPa levels minus the zonal wind at the 850 hPa level (smoothing parameter `n=6`).





(iii) `core_peak`: The latitude of the maximum of the zonal wind averaged between the 100 and 400 hPa levels (smoothing parameter `n=30`).

(iv) `core_max`: The latitude of the maximum of the zonal wind averaged between the 100 and 400 hPa levels (smoothing parameter `n=6`).

In all of the above methods, the latitude of the maximum is calculated poleward of 10° and equatorward of 60° for the `core` methods and equatorward of the latitude of the maximum of the zonal wind at the 850 hPa level in each hemisphere (i.e., equatorward of the eddy-driven jet, see below) for the `adjusted` methods. To reduce sensitivity to pressure-level spacing, vertical averages are pressure weighted (cf. Archer and Caldeira, 2008; Davis and Rosenlof, 2012). For all of the above methods, inputting the the value of `n` is optional and overrides the default values (i.e., 6 for `max` and 30 for `peak`).

## 3.5   EDJ – Eddy driven jet

The macroturbulent eddy momentum fluxes in midlatitudes, which drive the midlatitude jets, affect the zonal mean overturning circulation and therefore the tropical width (Kim and Lee, 2001; Schneider, 2006). Indeed, under some conditions, strong correlations are found between the positions of the EDJs and the width of the Hadley circulation, in particular in the SH during the summer months (Kang and Polvani, 2011). Since, in contrast to the STJs, the midlatitude EDJs are characterized

by relatively strong near-surface westerlies, EDJ-based metrics are generally calculated as the latitude of the maximum of near-surface westerlies (e.g., Woollings et al., 2010; Kang and Polvani, 2011; Davis and Birner, 2017).

**Methods**

To reduce grid dependence, it is common practice to fit a quadratic polynomial onto data from gridpoints surrounding the gridpoint of the maximum, and define the position of the EDJ as the latitude of the maximum of that polynomial (e.g., Kidston

and Gerber, 2010; Solomon et al., 2016). Davis and Birner (2017) use a linear interpolation of the gradient of the zonal mean zonal wind at 850 hPa to estimate the position of the EDJ. For consistency, the preferred methodology of the EDJ metric in TropD uses Eq. (1). For reference with previous studies, a generalized method based on a quadratic polynomial fit is also included.

**Comments on the code**

The EDJ metric is calculated in TropD using the syntax:

```
» [Ys Yn] = TropD_Metric_EDJ(U,lat,lev,method)
```

where `U(lat,lev)` denotes the zonal-mean wind. The methods are:

(i) `peak` (default): The latitude of the maximum of the zonal wind at the level closest to 850 hPa (smoothing parameter `n=30`).

(ii) `max`: The latitude of the maximum of the zonal wind at the level closest to 850 hPa (smoothing parameter `n=6`).





(iii) `fit`: The latitude of the maximum of a quadratic polynomial fit using `m` grid points on either side of the gridpoint of maximal zonal wind at the level closest to 850 hPa. The default value of `m` is 1.

The values of the smoothing parameter `n` in the `max` and `peak` methods ($n \geq 1$) and of the number of gridpoints on either side of the polynomial fit in the `fit` method ($m = 1, 2, 3 \ldots$) can be set as an input scalar after the method string; e.g.,

» `[Ys Yn] = TropD_Metric_EDJ(U,lat,lev,'max',n)`

or

» `[Ys Yn] = TropD_Metric_EDJ(U,lat,lev,'fit',m)`

In all of the above EDJ methods, the latitude of the maximum is calculated poleward of 15° and equatorward of 70° (i.e., slightly poleward of 60°, which is the poleward boundary in all of the other metrics).

**3.6 PE – Precipitation minus evaporation**

The subtropical dry zones lie at the latitude bands of the descending branches of the tropical meridional overturning circulation. The poleward edges of the subtropical dry zones can therefore be used as indices of the tropical width (e.g., Lu et al., 2007). The PE metric is calculated in TropD using the syntax:

» `[Ys Yn] = TropD_Metric_PE(PE,lat,method,Lat_Uncertainty)`

where `PE(lat)` denotes the zonal-mean precipitation minus evaporation field (Fig. 7a). The default and only available method is `zero_crossing` which calculates the zero-crossing latitude poleward of the subtropical minimum in `PE` and equatorward of 60°. The default value of the input parameter `Lat_Uncertainty` (optional), used by the function `TropD_Calculate_ZeroCrossing` as described above, is zero.

**3.7 UAS – Near-surface zonal wind**

The edge of the tropics is characterized by a transition from surface easterlies in the equatorward-flowing lower tropospheric branch of the Hadley circulation to surface westerlies in midlatitudes (Fig. 7b). [In steady state, the zonal-mean column-averaged zonal momentum flux divergence is balanced by surface drag. Therefore, the subtropical latitude where the zonal surface wind changes sign (i.e., where surface drag vanishes) also indicates the latitude where eddy and mean momentum flux divergences balance at higher levels (Held, 2000; Korty and Schneider, 2008).] Analyses of the zero-crossing latitude of surface

zonal wind are generally insensitive to the exact definition of the surface wind (e.g., the average wind 2m or 10m above surface, or the interpolated wind at the 1000 hPa level; Davis and Birner, 2017). Therefore, the UAS metric is calculated in TropD as the zero-crossing latitude of the zonal-mean near-surface zonal wind. The default and only available UAS metric method is `zero_crossing` which calculates the zero-crossing latitude poleward of the subtropical minimum in each hemisphere and equatorward of 60°. The default value of the input parameter `Lat_Uncertainty` (optional), used by the function

`TropD_Calculate_ZeroCrossing` as described above, is zero. The syntax for the UAS metric is:





```
» [Ys Yn] = TropD_Metric_UAS(U,lat,method,Lat_Uncertainty)
```

where `U(lat)` denotes the zonal-mean near-surface zonal wind (e.g., the wind 2m or 10m above the surface, or the wind at some level below 850 hPa).

### 3.8 PSL – maximum sea-level pressure

The subtropical high-pressure belts form along the descending branches of the tropical meridional overturning circulation. The latitude of maximum sea-level pressure may therefore serve as a tropical width indicator (Hu et al., 2011; Choi et al., 2014) (The use of sea-level pressure as opposed to surface pressure limits the influence of elevation over continents). In addition, sufficiently far from the equator, the geostrophically-balanced zonal wind changes sign where the meridional pressure gradient changes sign. Therefore, the latitude of maximum sea-level pressure lies near the latitude where the zonal-mean zonal surface

wind changes sign (i.e., it is closely related to the UAS metric; Choi et al., 2014), particularly in the SH (Fig. 7; Waugh et al., 2018). To reduce grid dependence, several studies have used procedures which rely on nonlinear interpolation to identify the position of the sea-level pressure maximum (e.g., Hu et al., 2011; Choi et al., 2014). For consistency, the methods for calculating the PSL metric in TropD are the same as for the EDJ metric:

(i) `max`: The latitude of the maximum of sea-level pressure (smoothing parameter `n=6`).

(ii) `peak` (default): The latitude of the maximum of sea-level pressure (smoothing parameter `n=30`).

The syntax for the PSL metric is:

```
» [Ys Yn] = TropD_Metric_PSL(PSL,lat,method)
```

where `PSL(lat)` denotes zonal-mean sea-level pressure.

In both the `max` and `peak` methods, the latitude of the maximum is calculated poleward of $15°$ and equatorward of $60°$. As

in the EDJ, STJ, and OLR metrics, the value of n can be set as an input scalar ($n \geq 1$) after the method string; e.g.,

```
» [Ys Yn] = TropD_Metric_PSL(PSL,lat,'max',n).
```

### 4   Method sensitivity analysis

We proceed with a method sensitivity analysis for the eight metrics implemented in TropD. For clarity, we use `[metric]:[method]` to denote metric and method.

### 4.1   Temporal averaging

It is important to note that, since the basic operators (max finding and zero crossing) applied in the metric calculations are not linear, the metric calculations do not commute in space and in time. This is illustrated in Fig. 8, where the PSI metric is calculated from monthly and annual means. The annual means are calculated using the TropD function `TropD_Calculate_Mon2Season` which calculates seasonal means from a monthly time series (see the example code and

the documentation in the code for syntax).



The annual means of `PSI:Psi_500` derived from monthly means of $v$ ($\langle$`PSI:Psi_500`$\rangle_{ANN}$, blue line) clearly differ from `PSI:Psi_500` derived from annual means of $v$ (`PSI:`$\langle$`Psi_500`$\rangle_{ANN}$, black line). The different seasonal averaging also yields slightly different decadal trends in the NH and SH ($0.27\pm0.21°$/decade for the annual-mean of the metric derived from monthly means vs. $0.12\pm0.24°$/decade for the metric derived from annual means of $v$ in the NH; and similarly, $-0.30\pm0.16°$/decade vs. $-0.34\pm0.18°$/decade in the SH; confidence bounds indicate 5% significance level using an F-test).

The agreement between dynamically consistent metrics is found to improve when these are derived from seasonal means as opposed to monthly means (Waugh et al., 2018). Similarly, the application of the zero-crossing uniqueness condition with `Lat_Uncertainty=10` yields 3 invalid results for monthly metric values in the NH but none for the metric values derived from annual means (Fig. 8a) or seasonal means (not shown). Therefore, for consistency in the analysis, we advocate that *metrics should be calculated from the seasonal means of the input field, as opposed to calculating the seasonal means of the metric*.

## 4.2 Grid dependence

To study the grid dependence of the various metric methods, we examine the relation of the inter-annual variability (the standard deviation of annual-mean values during 1979–2004) and the latitudinal grid spacing of the CMIP5 model output. Table 2 shows the inter-model correlation between the inter-annual variability and the latitudinal spacing for each of the metric methods. The grid dependence of the metric methods is generally higher in the SH, in particular for the near-surface metrics. This difference between the SH and NH may reflect smaller signal to noise ratio in the NH due to larger surface variability (i.e., it implies that grid dependence is more apparent in the SH, but not necessarily larger than in the NH). In addition, Davis and Birner (2016) find that (native) model resolution can affect eddy momentum and heat fluxes and therefore indirectly affect the large-scale circulation.

Given this sensitivity to grid spacing, method and model selection can play a critical role in reducing the uncertainty in analyses of the tropical width. For example, the inter-annual variability of the CMCC-CESM model, which has the lowest latitudinal resolution of the CMIP5 models considered here ($3.68°$, Table 1), is maximal or among the highest for most metrics, reflecting uncertainty in diagnosing the edge of the tropics rather than physical variability; excluding low-resolution models from analyses of the tropical width is therefore one way of reducing uncertainty. Likewise, in analyses of the tropical width that use data ensembles with large grid variance, uncertainty can be reduced by using metrics which are not sensitive to grid spacing (e.g., the PSI metric).

## 4.3 Sensitivity to the smoothing parameter `n`

In the presence of random noise, the `max` method which uses moderate smoothing (`n=6`), and the `peak` method which uses weak smoothing (`n=30`) and is therefore more sensitive to noise, can produce significantly different estimates of the latitude of the maximum (Fig. 3). However, in both observations and simulations, it is not clear whether differences between the `max` and `peak` methods reflect grid dependence, sensitivity to noise, or a realistic quantification of physical variability. In other words, the optimal smoothing level, which minimizes grid dependence and noise while retaining the relevant physical properties of the analyzed variable, is not known.



To demonstrate the sensitivity of the various methods to the smoothing parameter n, the sensitivity of the `STJ:core` and `STJ:adjusted` methods, and of the `TPB:max_gradient` and `TPB:max_potemp` methods to n is analyzed in Figs. 9 and 10. Additional information on the sensitivity to n of the inter-annual variability and decadal trends in these metrics is provided in the Supplementary Material (SM, Figs. S1–S4). Consistent with Fig. 3, inter-model spread generally increases with
n for `STJ:core`, presumably due to increased sensitivity to noise, but not for `STJ:adjusted`. The standard deviation of inter-model spread is also significantly greater in the SH for `STJ:core` (greater than 8° for n>10, Fig. 9a) compared with `STJ:adjusted` (∼2° for n≥6), but is approximately the same for both methods in the NH (∼4° for n≥6; Fig. 9b,d). This suggests that the differences between the two methods in the SH are due to the stronger signal of the midlatitude jet in the SH, which is successfully removed in the `STJ:adjusted` method (Fig. 6).

Due to the sharp gradient in the tropopause height at the tropopause break (Fig. 4), the mean position of the `TPB:max_gradient` method (which identifies the latitude of maximal meridional gradient; Sec. 3.2) is insensitive to the smoothing parameter in both hemispheres for n≥ 6 (Fig. 10a,b). However, the inter-model spread in inter-annual variability and decadal trends generally increase with n (Fig. S3). Therefore, the default smoothing parameter for this method is set to n=6 (i.e., the `max` method). In contrast, the `TPB:max_potemp` method (which identifies the latitude of maximal difference in the
potential temperature between the tropopause and the surface; Sec. 3.2) is insensitive to the smoothing parameter for n≥20 (Figs. 10c-d, S4). The default smoothing parameter for this method is therefore set to n=30 (i.e., the `peak` method). Likewise, since for most of the metrics the optimal smoothing level is not known, to reduce method dependence on external parameters, the `peak` method (i.e., weak smoothing) is preferred as the default method for finding the latitude of the maximum.

### 4.4 Inter-method analysis

The mean values of the default methods in each of the eight metrics implemented in TropD are shown in Fig. 11. For simplicity, our inter-method analysis focuses on the mean values of the various metrics, derived from annual-means, for the period 1979–2004. Additional method sensitivity analysis for the inter-annual variability and the decadal trends of the eight metrics is provided in the SM (Figs. S5–S9).

Fig. 12 shows the five available methods of the PSI metric derived from annual-mean values of the zonal mean meridional
wind. The five PSI methods vary consistently across models (Fig. 12c,d). In addition, the differences between the five PSI methods per model (∼ 1° on average) are generally smaller than the inter-model spread (∼ 2° standard deviation). Therefore, given the prevalence of the `Psi_500` method in studies of the tropical width, this method is set as the default method for the PSI metric in TropD.

Similar candle plots for the TPB and OLR metrics are shown in Figs. 13 and 14, for the STJ and EDJ metrics in Fig.
15, and for the PE, UAS and PSL metrics in Fig. 16. As for the PSI metric, the default method for each metric is shown in bolded text. CMIP5 models generally underestimate mean metric values, interannual variability, and trend in both PSI and TPB metrics when compared with ERAI (Figs. 12, 13, S5 and S6), but are generally consistent with ERAI values for the other metrics (Figs. 14–16, S7–S9). As reported in previous studies, inconsistencies between reanalyses and models during this period are attributable to internal variability, variations in the response to anthropogenic forcing, and the unknown reliability of



the reanalyses (e.g., Adam et al., 2014; Garfinkel et al., 2015; Mantsis et al., 2017). However, for all of the metrics, the relative variations between the methods in both ERAI and the CMIP5 models are generally the same. This reinforces our assertion that *uncertainty in analyses of tropical width variations can be reduced by applying consistent methodologies across datasets*.

Figure 4 shows the tropopause height and the difference in the potential temperature between the tropopause and the surface during the decades beginning in 1979 and 1995. Consistent with theory in a warming climate (e.g., Levine and Schneider, 2015), both the tropopause height and the potential temperature difference increase in the tropics during 1979–2004. However, the changes in the tropopause height near the tropopause break differ substantially between hemispheres, while the change in the potential temperature difference is uniform across the tropics. These qualitative differences in the profiles of the tropopause height and the potential temperature account for the differences in the `TPB:max_gradient` and `TPB:max_potemp` methods (Figs. 13, S3-S4, S6) and may lead to differing estimates of tropical width variations. In addition, the mean value, interannual variability and decadal trends of the `TPB:cutoff` method are sensitive to the cutoff parameter (Figs. 13, S6). The variance across models generally increases as the cutoff value nears the maximal value of the tropopause height (∼16km) in the tropics, which goes along with reduced meridional gradients of the tropopause and hence increased sensitivity to the cutoff parameter. Due to its simplicity and its direct association with the tropopause height, `max_gradient` is set as the default method for the TPB metric.

A similar sensitivity to subjective cutoff parameters is seen in the OLR metric (i.e., the `cutoff`, `20W`, and `10Perc` methods, Figs. 14, S7). Due to uncertainty in observations, the reliability of the zonal-mean OLR in CMIP5 models is poorly known (e.g., Fig. 5; Stephens et al., 2012), but OLR anomalies in CMIP5 models are generally well correlated with observations (Smith et al., 2015). However, given the strong sensitivity of the OLR metric to subjective cutoff parameters, and since decadal changes in OLR are generally smaller than the inter-model spread, estimates of recent tropical widening using this metric are highly uncertain (Davis and Rosenlof, 2012; Waugh et al., 2018). The `250W` method, which is the most commonly used OLR metric (e.g., Hu and Fu, 2007; Johanson and Fu, 2009; Davis and Rosenlof, 2012), is set as the default OLR method.

The subtraction of the 850 hPa wind from the upper-level zonal wind distinguishes the signal of the subtropical jet from that of the eddy-driven jet (Fig. 6; Davis and Birner, 2016), accounting for the strong differences between the `STJ:core` and `STJ:adjusted` methods (Fig. 15). In addition, since below the STJ the surface zonal wind is expected to vanish, the subtraction of near-surface wind is not expected to strongly affect the signal of the STJ near the edge of the Hadley circulation. Since, in addition, the subtraction of the 850 hPa wind reduces inter-model spread (Figs. 6, 15), the `adjusted_peak` method is set as the default method for the STJ metric. As mentioned above, because the optimal smoothing level is not known, the `peak` method is set as the default method for finding the latitude of the maximum for both the the STJ and EDJ metrics.

The subtropical meridional profiles of the upper- and lower-level zonal wind (Fig. 6) and the sea-level pressure (Fig. 7c) are significantly more flat in the NH relative to the SH, making the STJ, EDJ, and PSL metrics more sensitive to noise in the NH. Therefore, using different smoothing levels for the SH and NH metrics may be advisable in some cases (e.g., using `max` in the NH and `peak` in the SH, as in Waugh et al., 2018). Similarly, due to continental effects, variance near the surface in the NH subtropics is generally greater than in the SH. This can lead to differences between hemispheres when the zero-crossing uniqueness criteria is used (e.g., Fig. 8). Therefore, using different values of the `Lat_Uncertainty` parameter for NH and



SH metrics may be advisable in some cases for the near-surface metrics UAS and PE, as well as for PSI. For the UAS metric, the above holds even when the input wind is the zonal-wind at the 850 hPa level (Fig. 6), which produces more equatorward UAS metric values, but varies consistently across models with UAS values derived from the surface wind (Fig. 16).

## 5    Conclusions

The TropD software package provides methodologies for eight commonly-used metrics of the tropical width. TropD is designed to reduce, or aid in the assessment of, the methodological component of the uncertainty in studies of tropical width variations by:

1. Compiling the relevant methods for each metric category

2. Reducing all of the calculations in the metric methods to two basic operations: (i) finding the latitude of the zero crossing, or (ii) finding the latitude of the maximum

3. Providing functions for calculating the meridional mass streamfunction and the tropopause height according to generally accepted guidelines

4. Providing consistent methods for implementing threshold criteria

5. Using consistent smoothing across methods

6. Using consistent meridional limits for the various metrics

In addition, TropD allows flexibility in the input parameters (e.g., in the cutoff and smoothing parameters) which enables consistent sensitivity testing of the metrics in various datasets, as well as testing new methods.

Our method sensitivity analysis highlights the importance of differentiating between variations which arise from parameter choices and inconsistent resolutions across datasets, as opposed to differences which arise from the representation of physical processes in the datasets. The analysis suggests that careful use of the metric methods can reduce some of the spurious uncertainty. For example, using different metric methods for each hemisphere can minimize the spurious uncertainty seen in inter-model variations of the surface metrics (Fig. 16) and the grid sensitivity of the EDJ metric (Table 2). Similarly, spurious uncertainty can be significantly reduced by proper method selection, which can minimize undesired effects such as temporal averaging biases, grid dependence, sensitivity to noise, and sensitivity to threshold criteria.

Based on our inter-method analysis, the default methods and parameters for each metric category are optimized for the present climate. Nevertheless, the TropD code can be easily adapted for studies of past climates or perturbations of the present climate, which, as in studies of recent tropical width variations, suffer from unknown spurious uncertainty. Similarly, elements of the TropD code can be applied to a wide range of studies beyond calculations of the tropical width (e.g., the position of the intertropical convergence zone, tropopause height variations, and circulation intensity variations) where the use of standardized methodology can reduce spurious uncertainty.





*Code and data availability.* The TropD software, reference pre-calculated metrics, and reference fields are freely available via *www.Zenodo.org*, doi:10.5281/zenodo.1157043, and at *www.oriadam.info*

.

*Author contributions.* The MATLAB Code was written by Ori Adam, with algorithmic contributions from the co-authors.

5   *Competing interests.* The authors declare that they have no conflict of interest.

*Acknowledgements.* This work is part of the collaborative efforts of the International Space Science Institute (ISSI) Tropical Width Diagnostics Intercomparison Project and the U.S. Climate Variability and Predictability Program (US CLIVAR) Changing Width of the Tropical Belt Working Group. The authors thank the members of these groups as well as the ISSI and US CLIVAR offices and sponsoring agencies (ESA, Swiss Confederation, Swiss Academy of Sciences, University of Bern, NASA, NOAA, NSF, and DOE) for their support. We acknowledge
10   the World Climate Research Programme's Working Group on Coupled Modelling, which is responsible for CMIP, and we thank the climate modeling groups for producing and making available their model output. Ori Adam acknowledges support by the Israeli Science Foundation grant 1185/17.



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



**Table 1.** ERAI and CMIP5 models' affiliations and the horizontal resolution of the analyzed data (lon$^\circ \times$lat$^\circ$). The first ensemble member ('r1i1p1') is used from each CMIP5 model.

| Model | Affiliation | Output resolution |
|---|---|---|
| ERAI | ECMWF interim reanalysis | $1.50 \times 1.50$ |
| ACCESS1-0 | CSIRO, and Bureau of Meteorology, Australia | $1.88 \times 1.25$ |
| ACCESS1-3 | CSIRO, and Bureau of Meteorology, Australia | $1.88 \times 1.25$ |
| BNU-ESM | Beijing Normal University (BNU), China | $2.81 \times 2.77$ |
| CMCC-CESM | Euro-Mediterranean Center on Climate Change (CMCC), Italy | $3.75 \times 3.68$ |
| CNRM-CM5 | CNRM and CERFACS, France | $1.41 \times 1.39$ |
| CNRM-CM5-2 | CNRM and CERFACS, France | $1.41 \times 1.39$ |
| CSIRO-Mk3-6-0 | CSIRO, and Bureau of Meteorology, Australia | $1.88 \times 1.85$ |
| CanESM2 | CCCma, Canada | $2.81 \times 2.77$ |
| GFDL-CM3 | NOAA GFDL, USA | $2.50 \times 2.00$ |
| GFDL-ESM2G | NOAA GFDL, USA | $2.50 \times 1.52$ |
| GFDL-ESM2M | NOAA GFDL, USA | $2.50 \times 1.52$ |
| GISS-E2-H | NASA, USA | $2.50 \times 2.00$ |
| GISS-E2-H-CC | NASA, USA | $2.50 \times 2.00$ |
| GISS-E2-R | NASA, USA | $2.50 \times 2.00$ |
| GISS-E2-R-CC | NASA, USA | $2.50 \times 2.00$ |
| HadCM3 | Met Office Hadley Centre, UK | $3.75 \times 2.50$ |
| HadGEM2-AO | Met Office Hadley Centre, UK | $1.88 \times 1.25$ |
| IPSL-CM5A-LR | Institut Pierre Simon Laplace (IPSL), France | $3.75 \times 1.89$ |
| IPSL-CM5A-MR | Institut Pierre Simon Laplace (IPSL), France | $2.50 \times 1.27$ |
| IPSL-CM5B-LR | Institut Pierre Simon Laplace (IPSL), France | $3.75 \times 1.89$ |
| MIROC-ESM | JAMSTEC, AORI, and NIES, Japan | $2.81 \times 2.77$ |
| MIROC-ESM-CHEM | JAMSTEC, AORI, and NIES, Japan | $2.81 \times 2.77$ |
| MIROC4h | JAMSTEC, AORI, and NIES, Japan | $0.56 \times 0.56$ |
| MIROC5 | JAMSTEC, AORI, and NIES, Japan | $1.41 \times 1.39$ |
| MPI-ESM-LR | Max Planck Institute for Meteorology, Germany | $1.88 \times 1.85$ |
| MPI-ESM-MR | Max Planck Institute for Meteorology, Germany | $1.88 \times 1.85$ |
| MPI-ESM-P | Max Planck Institute for Meteorology, Germany | $1.88 \times 1.85$ |
| MRI-ESM1 | Meteorological Research Institute (MRI), Japan | $1.12 \times 1.11$ |
| MRI-CGCM3 | Meteorological Research Institute (MRI), Japan | $2.12 \times 1.11$ |
| NorESM1-M | Norwegian Climate Center, Norway | $2.50 \times 1.89$ |
| NorESM1-ME | Norwegian Climate Center, Norway | $2.50 \times 1.89$ |
| bcc-csm1-1 | Beijing Climate Center (BCC), China | $2.81 \times 2.77$ |
| bcc-csm1-1-m | Beijing Climate Center (BCC), China | $1.12 \times 1.11$ |
| inmcm4 | Institute for Numerical Mathematics (INM), Russia | $2.00 \times 1.50$ |





**Table 2.** The Pearson coefficient of inter-model correlation between the inter-annual variability (defined as the standard deviation of annual-mean values for 1979–2004) of the metric and the latitudinal spacing of the model output in the 34 CMIP5 models. Default methods and statistically significant correlations ($p<0.05$) are bolded.

| Method | SH | NH |
|---|---|---|
| **PSI:Psi_500** | 0.04 | 0.19 |
| PSI:Psi_500_10Perc | -0.03 | 0.20 |
| PSI:Psi_Levels | 0.02 | 0.16 |
| PSI:Psi_500_Int | -0.19 | 0.20 |
| PSI:Psi_Int | 0.01 | 0.04 |
| **TPB:max_gradient** | 0.32 | -0.22 |
| TPB:max_potemp | 0.06 | 0.10 |
| TPB:cutoff=14,500 | -0.07 | -0.05 |
| TPB:cutoff=15,000 | 0.10 | 0.04 |
| TPB:cutoff=15,500 | -0.10 | -0.12 |
| **OLR:250W** | 0.05 | 0.10 |
| OLR:cutoff=240 | 0.11 | 0.10 |
| OLR:20W | 0.29 | 0.30 |
| OLR:10Perc | 0.27 | 0.20 |
| **STJ:adjusted_peak** | 0.30 | 0.04 |
| STJ:adjusted_max | 0.23 | -0.08 |
| STJ:core_peak | 0.13 | -0.03 |
| STJ:core_max | 0.20 | 0.01 |
| **EDJ:peak** | 0.27 | -0.16 |
| EDJ:max | **0.38** | 0.04 |
| EDJ:fit | 0.21 | -0.25 |
| **PE:zero_crossing** | 0.12 | -0.04 |
| **UAS:zero_crossing** | **0.50** | 0.20 |
| **UAS:zero_crossing**@850hPa | **0.50** | 0.09 |
| PSL:max | **0.54** | 0.12 |
| **PSL:peak** | **0.49** | 0.19 |





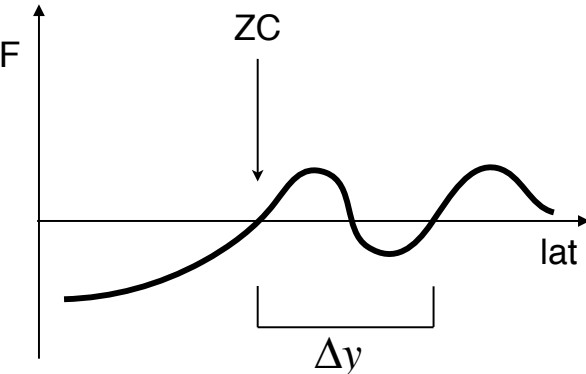

**Figure 1.** A depiction of the latitude of the zero crossing `ZC` of some field `F` along the interval `lat`. The distance to the nearest zero crossing with the same sign change is denoted by $\Delta y$.

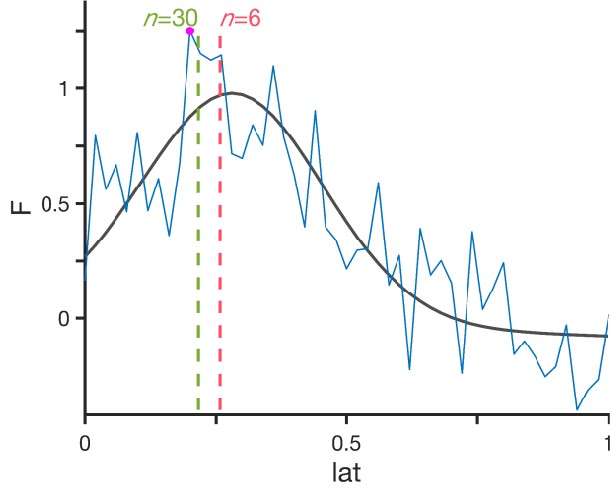

**Figure 2.** Example of the latitude of the maximum (`Ymax`) of a field `F`, calculated using `TropD_Calculate_MaxLat(F,lat,n)`. `F` is given as some skewed Gaussian function (black) with random noise (blue) on a discretized grid with a random resolution. The results for `n=6` (red) and `n=30` (green) are indicated by vertical lines. The absolute maximum is indicated by a dot (magenta).




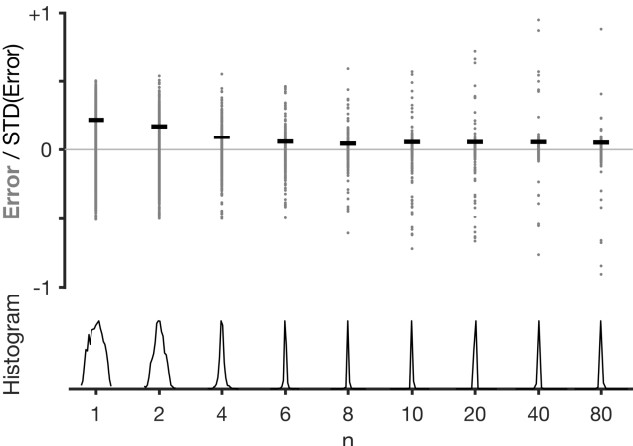

**Figure 3.** The dependence on `n` of the error distribution of $\phi_{max}$ calculated using Eq. (1) in a representative sample of 100 randomized skewed Gaussian functions such as the one shown in Fig. 2. The error (gray dots) is defined as the difference between $\phi_{max}$ and the latitude of the maximum of the smooth Gaussian function (black line in Fig. 2). Standard deviations of the error (STD(Error), horizontal lines) and histograms (normalized between 0 and 1) are shown for the error distributions of the sample for each `n`.

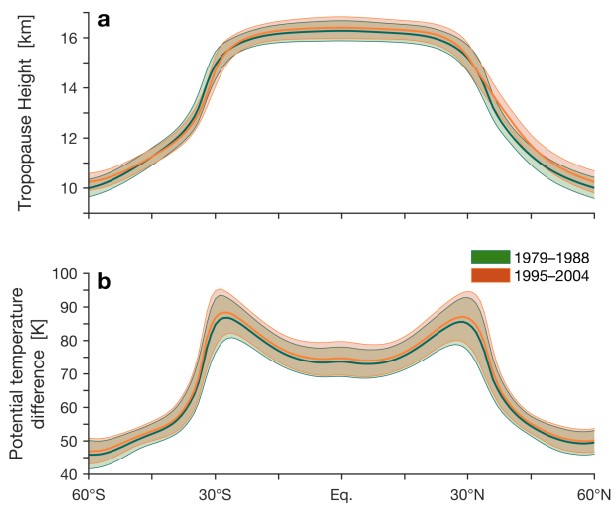

**Figure 4.** The mean tropopause height (**a**) and the difference in the potential temperature between the tropopause level and the surface (**b**) during the decades beginning in 1979 (green) and 1995 (orange) in CMIP5 models. The shading indicates $\pm 1$ standard deviation of inter-model spread. The calculations are derived from monthly means of the temperature field.





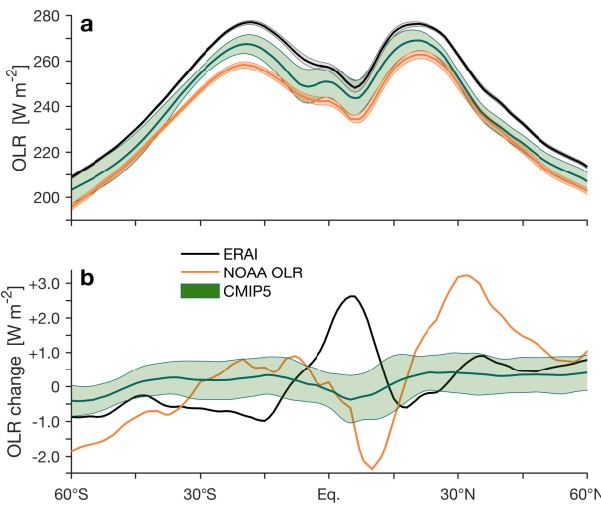

**Figure 5.** Outgoing longwave radiation (OLR) at the top of the atmosphere in CMIP5 models (green), ERAI (black) and the National Oceanic and Atmospheric Administration (NOAA) interpolated OLR dataset (orange, Liebmann and Smith, 1996). (**a**) The mean OLR during 1979–2004. (**b**) The difference in OLR between the decades beginning in 1995 and 1979. The shading indicates $\pm$ 1 standard deviation of inter-model spread for CMIP5 models and of inter-annual variations for ERAI and the NOAA interpolated OLR.

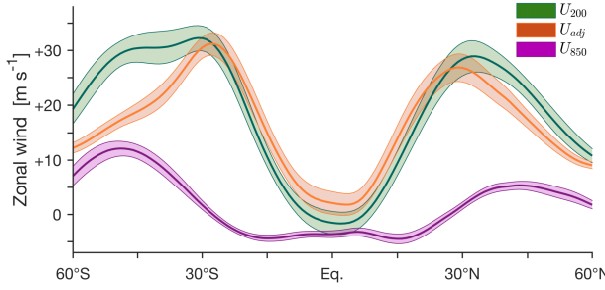

**Figure 6.** The zonally averaged annual-mean zonal wind at the 200 and 850 hPa levels ($U_{200}$, green, and $U_{850}$, purple) and the difference between $U_{200}$ and $U_{850}$ ($U_{adj}$, orange). Data taken from CMIP5 models for 1979–2004. The shading indicates $\pm$ 1 standard deviation of inter-model spread.





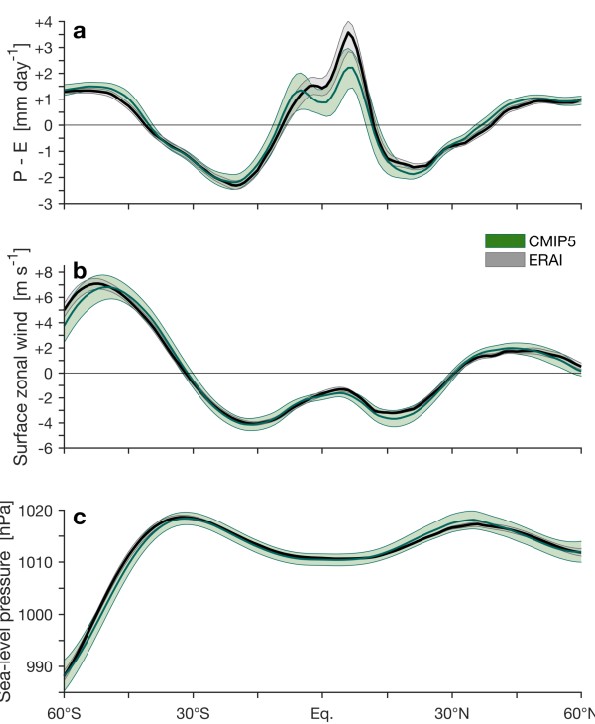

**Figure 7.** Zonally averaged annual-mean values of: (**a**) precipitation minus evaporation (P-E), (**b**) surface zonal wind, and (**c**) sea-level pressure in CMIP5 models (green) and ERAI (black) for 1979–2004. The shading indicates ± 1 standard deviation of inter-model spread for CMIP5 models and of inter-annual variations for ERAI.





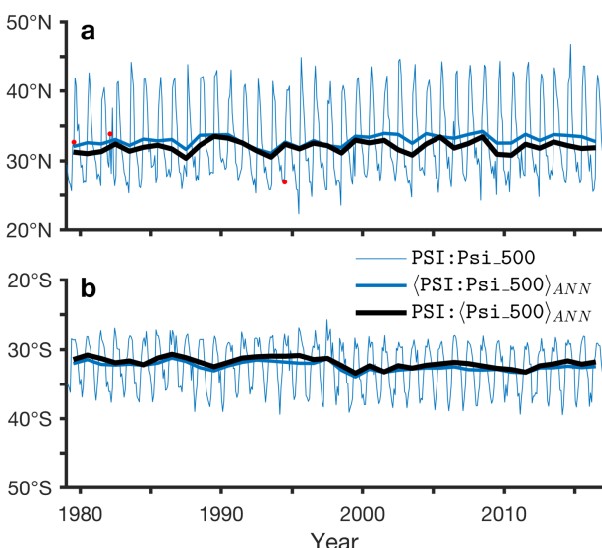

**Figure 8.** Time series of the PSI metric for the `Psi_500` method (default) in the NH (**a**) and SH (**b**). Shown are monthly values (thin blue), annual means of the monthly metric values ($\langle$`PSI:Psi_500`$\rangle_{ANN}$, thick blue), and metric values derived from the annual-mean meridional wind (`PSI:`$\langle$`Psi_500`$\rangle_{ANN}$, black). Data taken from ERAI for 1979–2016. The default value of the input parameter `Lat_Uncertainty` is zero. Monthly values with invalid results for `Lat_Uncertainty=10` are marked by red dots (3 in the NH and none in the SH).





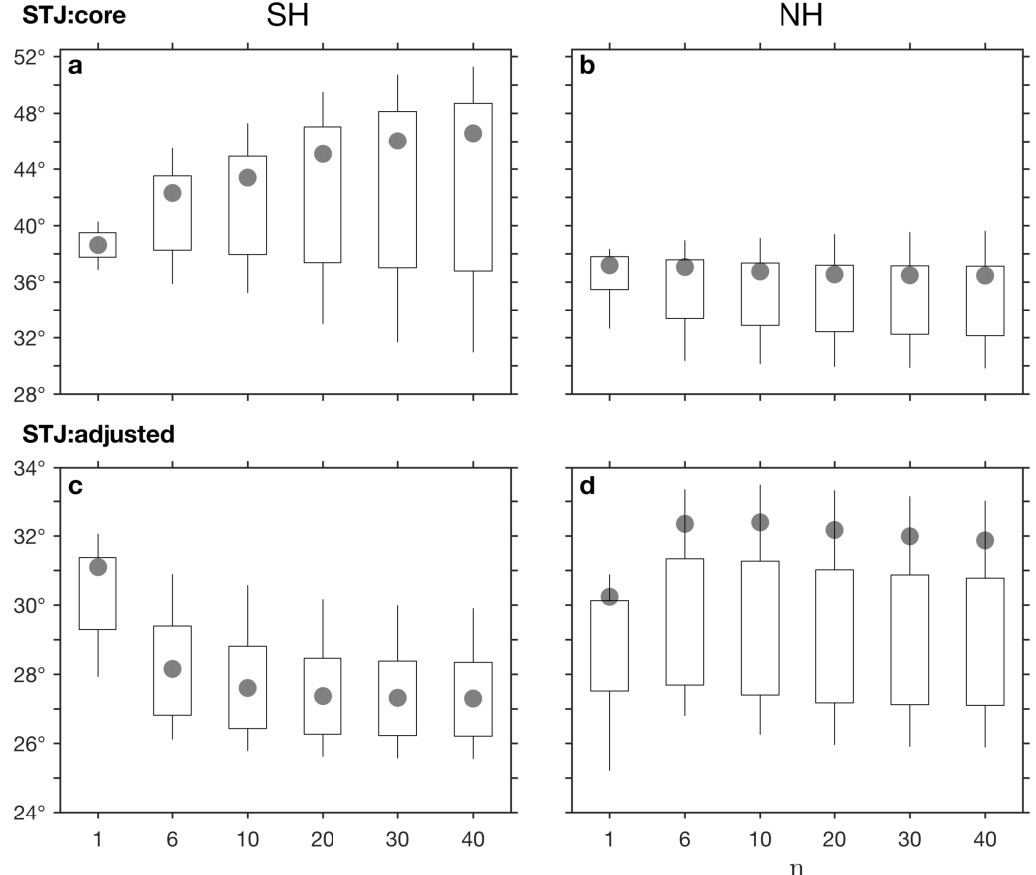

**Figure 9.** The dependence of the mean metric latitude on the smoothing parameter n for the `STJ:core` and `STJ:adjusted` methods, derived from annual means of the zonal-mean zonal wind in the SH (left) and NH (right), for the period 1979–2004. For simplicity, positive values are used for both the NH and SH. Candle boxes show mean $\pm$ 1 standard deviation of CMIP5 inter-model spread. Candle wicks show maximal and minimal CMIP5 model values. ERAI values are shown in gray dots.





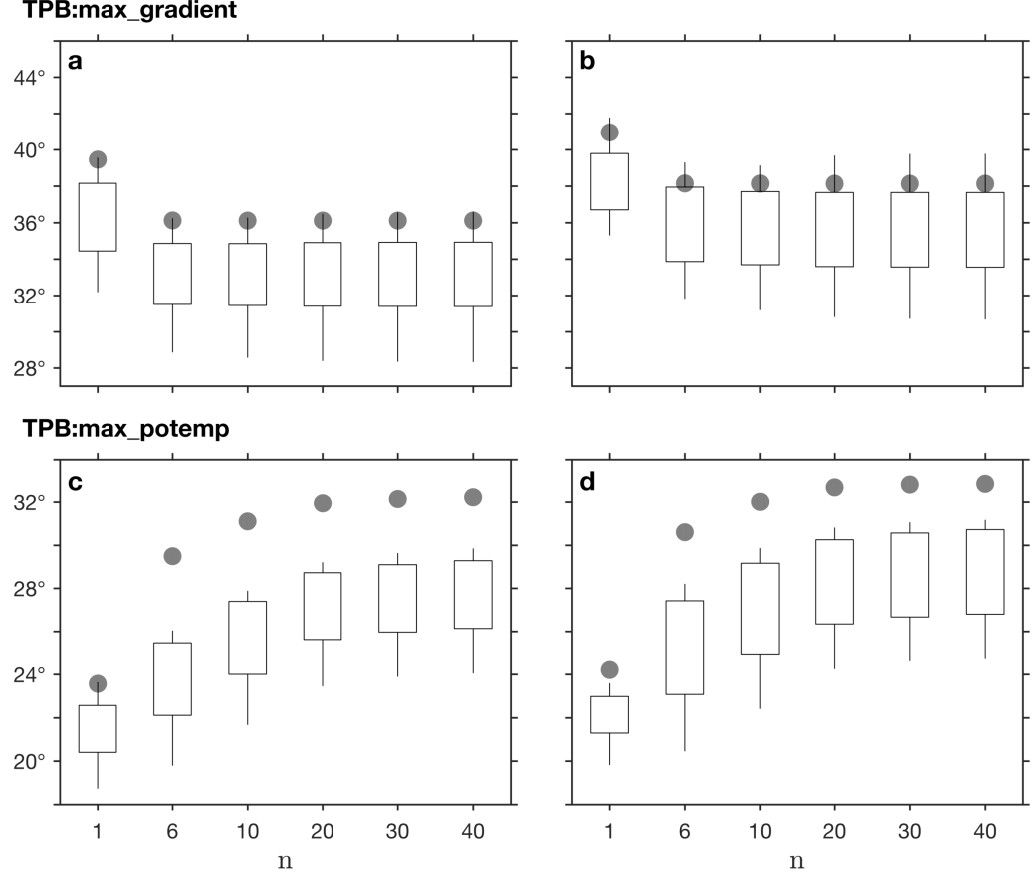

**Figure 10.** As in Fig. 9 for the dependence of `TPB:max_gradient` and `TPB:max_potemp` on the smoothing parameter n.



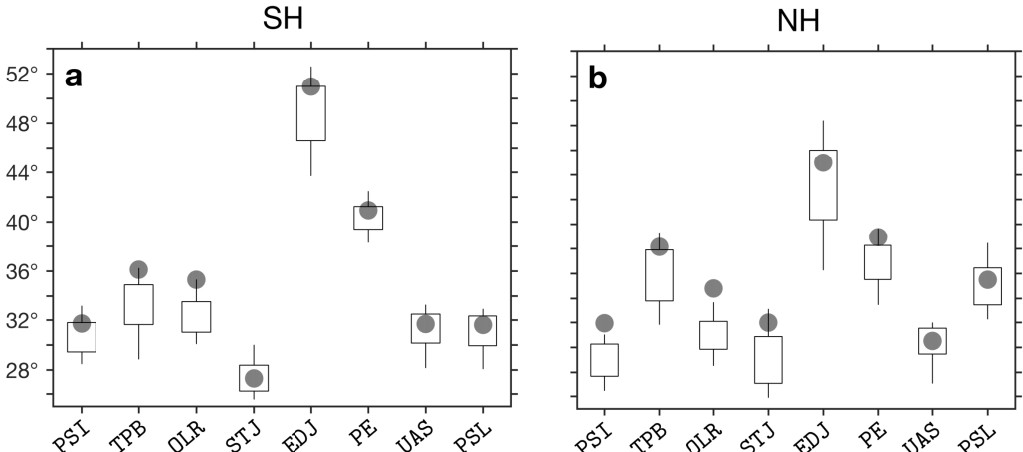

**Figure 11.** The mean SH (**a**) and NH (**b**) latitudes of the default methods in each of the eight metrics, derived from annual means, for the period 1979–2004. Positive values are used for both the NH and SH. Candle boxes show mean $\pm$ 1 standard deviation of CMIP5 inter-model spread. Candle wicks show maximal and minimal CMIP5 model values. ERAI values are shown in gray dots. The input parameter `Lat_Uncertainty` is set to zero (default) in all of the relevant methods.




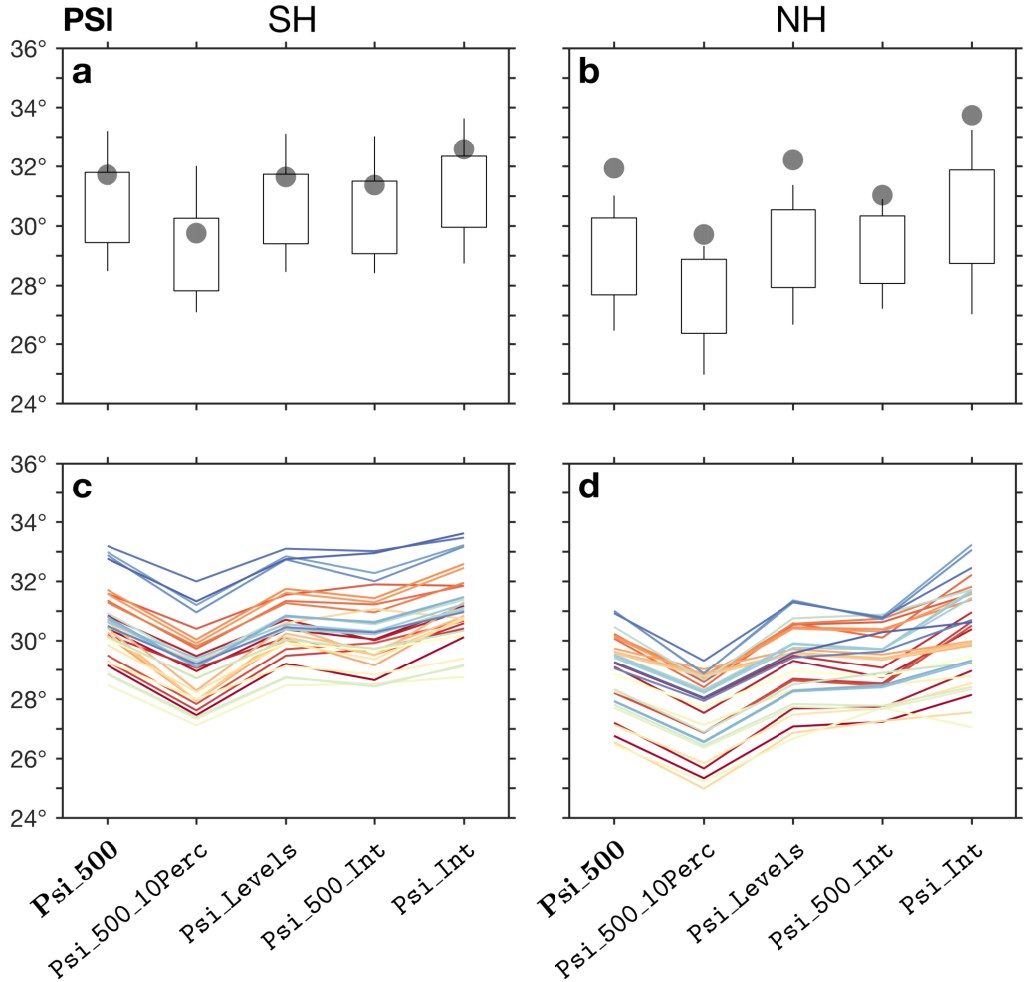

**Figure 12.** The mean latitude for the 5 available PSI metric methods, derived from annual means of the zonal-mean meridional wind in the SH (**a,c**) and NH (**b,d**), for the period 1979–2004. The default method (`Psi_500`) is bolded. Positive values are used for both the NH and SH. In panels **a** and **b**, candle boxes show mean ± 1 standard deviation of CMIP5 inter-model spread; candle wicks show maximal and minimal CMIP5 model values. ERAI values are shown in gray dots. In panels **c** and **d**, values for each model are shown in different colors. The input parameter `Lat_Uncertainty` is set to zero (default) in all of the methods.




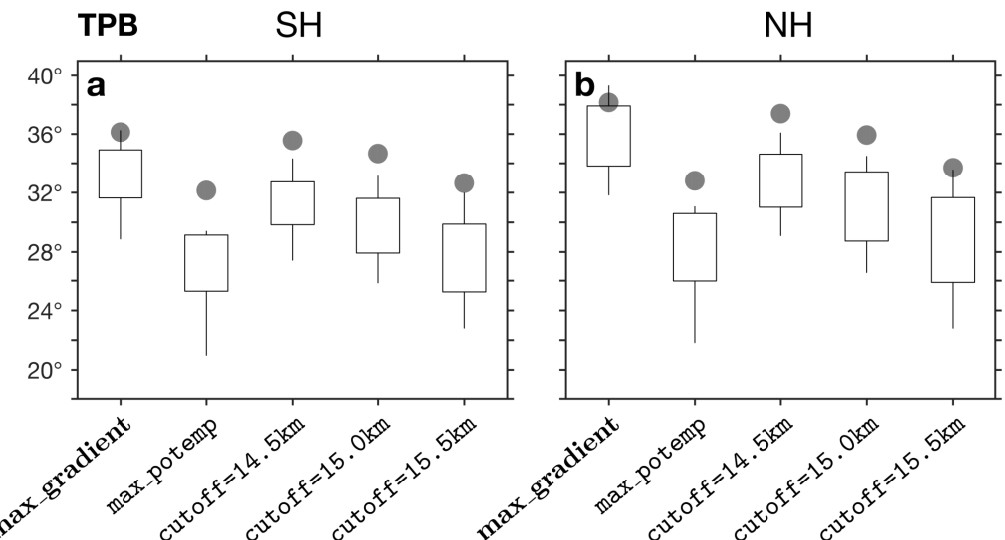

**Figure 13.** The mean latitude for the `max_gradient`, `max_potemp`, and `cutoff` methods of the TPB metric, as in Fig. 12a,b. For the `cutoff` method, geopotential height cutoff values of 14500m, 15000m (default) and 15500m are shown. For `Cutoff`=15,500 and for `max_potemp`, three of the models produced unrealistic results and were removed from the analysis.




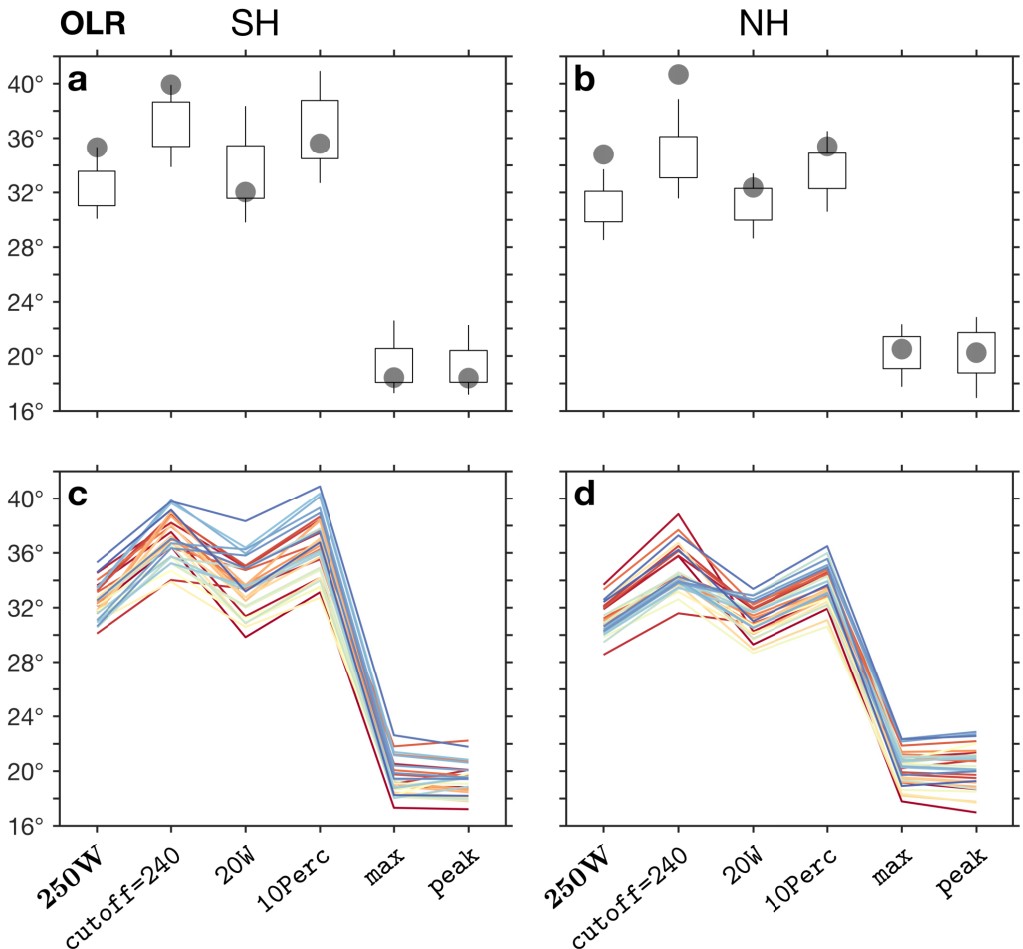

**Figure 14.** As in Fig. 12 for the OLR metric methods `250W`, `cutoff` (with `Cutoff=240`), `20W`, `10Perc`, `max`, and `peak`.



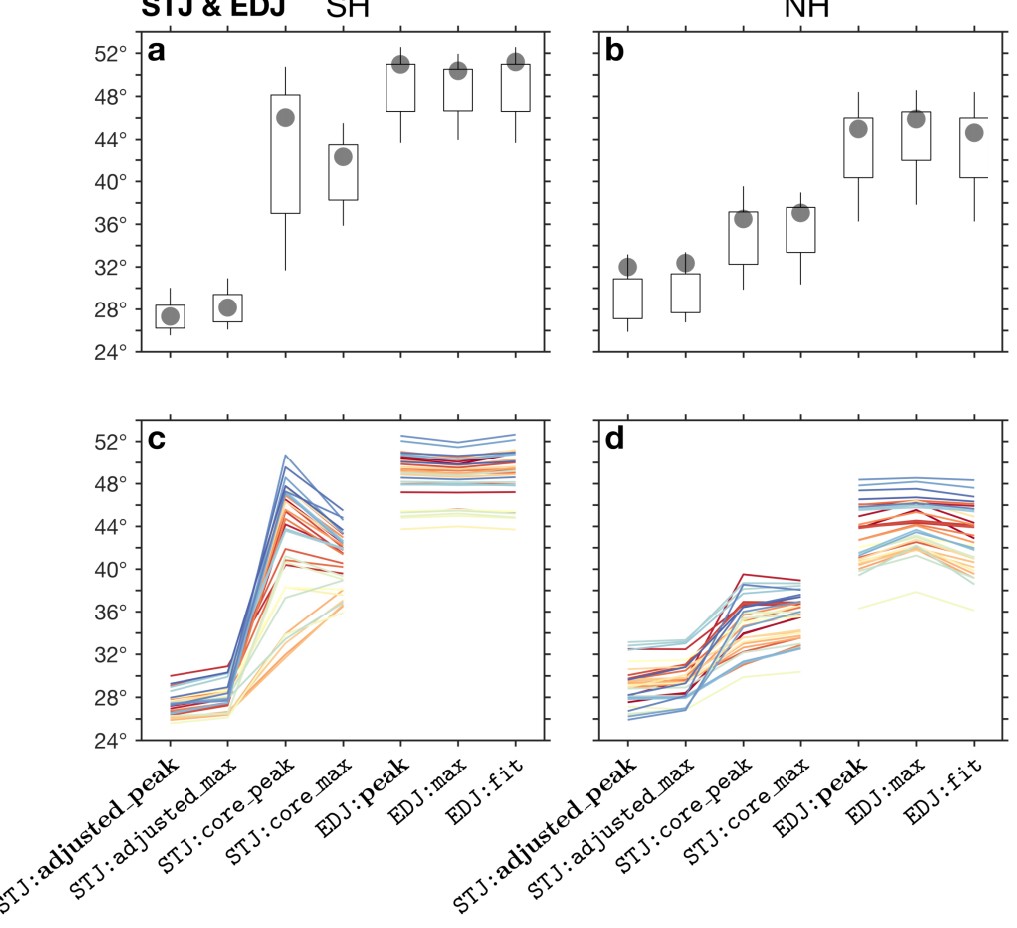

**Figure 15.** As in Fig. 12 for the STJ and EDJ metrics.





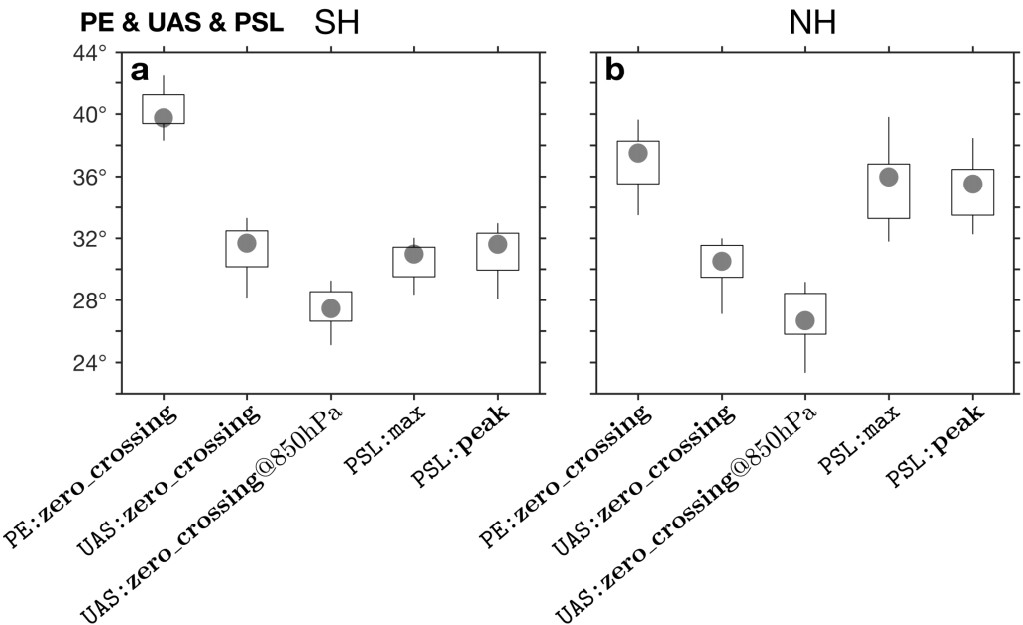

**Figure 16.** As in Fig. 12a,b for `PE:zero_crossing`, `UAS:zero_crossing` using the zonal wind both at the surface and at the 850 hPa level, `PSL:max`, and `PSL:peak`.