# Peer review of "The TropD software package: Standardized methods for calculating Tropical Width Diagnostics"

_Geoscientific Model Development, 2018_

## Short Comment (SC1) · 9 Jul 2018

Dear authors,

in my role as Executive editor of GMD, I would like to bring to your attention our Editorial version 1.1:

http://www.geosci-model-dev.net/8/3487/2015/gmd-8-3487-2015.html

This highlights some requirements of papers published in GMD, which is also available on the GMD website in the 'Manuscript Types' section:

http://www.geoscientific-model-development.net/submission/manuscript_types.html

In particular, please note that for your paper, the following requirement has not been

met in the Discussions paper:

- "The main paper must give the model name and version number (or other unique identifier) in the title."

Please add a version number for the software package TropD in the title upon your revised submission to GMD.

Yours,

Astrid Kerkweg
* * *

---

## Referee Comment (RC1) · Anonymous Referee #1 · 27 Jul 2018

This work presents a framework in MATLAB programming language to calculate the tropical width using the eight most common methods and metrics for this purpose. The package of code called TropD includes not only the source files where all methods covered in the article were implemented, but also includes examples, data and documentation.

TropD is provided under an open source licenses, so it can be used and modified without restrictions. In this way, users can adapt it to their own studies, making a significant contribution to the scientific community.

The article clearly explains each method implemented in the code package TropD. Each important function is implemented in a separate file with multiple options for its execution and its interface is properly documented. An example code and validation

data are included to evaluate the use of TropD.

No significant shortcomings were found in the implementation of this code package. The source code is clear and well organized. All variables are documented and the comments included in the code make this code easy to understand. Therefore, I have no suggestions for improving the code package presented in this work.

———————————————————

---

## Referee Comment (RC2) · Anonymous Referee #2 · 21 Aug 2018

This paper describes a software package that implements eight commonly-used methodologies for the calculation of metrics of the tropical width. A set of optional parameters is made available for some methodologies. The code is written in the widely used MATLAB programming language, and its purpose is to provide standardized calculation methodologies for commonly used diagnostics of the tropical width, helping in the comparison of the results from different studies.

The paper is well written and the software functions well described. A sensitivity analysis for the different methodologies is presented to justify the choice of the default options.

Because the basic operators applied in the metric calculations are nonlinear, the metric calculations do not commute in space and in time. The authors discussed this in

the manuscript. However, there is another effect of nonlinearity that the authors did not discuss. The input data for the software are zonal-mean variables in a latitude pressure grid. For example, in the TBP method, the zonal-mean tropopause is calculated by applying the WMO lapse rate criteria to the isobaric zonal mean temperature. I wonder if the results will be the same when the zonal mean tropopause parameters are obtained by zonally averaging the respective values calculated at each longitude.

I have more two minor comments:

pg. 3, line 16: "(ii) In cases where multiple zero-crossing latitudes exist, the first zero crossing along the input interval is chosen."

Because the data grid is ordered from the South Pole to the North Pole, the first zero-crossing latitude in the SH is far from the equator than the first zero-crossing latitude in the NH. Is it right?

Fig. 9: It will be useful to call the attention for the different vertical scales used in Figs. 9a,b and 9c,d.

---

## Author Comment (AC1) · 2 Sep 2018

A version identifier was added to the title (v1)

———————————————

---

## Author Comment (AC2) · 2 Sep 2018

We thank the reviewer for his comments.

1. With regards to the differences between applying the metric functions on zonal-mean fields vs. zonally averaging the metric functions applied per longitude. Indeed, as the reviewer suggests, for some metrics, the metric derived from a zonal-mean field may differ from the zonal average of the metric applied per longitude. However, as stated in Section 2.1, TropD is designed to be applied on zonal-mean fields. This is because the methodologies for zonally varying indices of the tropical width require more research, potentially followed by modifications of the code and methods. We may address this issue and incorporate methodologies for zonally-varying metrics of the tropical width in

later versions of TropD.

2. A clarification note was added to section 2.2 (latitude of zero crossing). The comment reads: "The metric functions described below automatically order the input interval lat such that the first latitude of zero crossing in each hemisphere corresponds to the most equatorward zero crossing."

3. The caption of Fig. 9 was edited to note the differences between the vertical scales of the STJ:core and STJ:adjusted methods
* * *

---

## Author Comment (AC3) · 2 Sep 2018

We thank the reviewer for his comments.

---

## Author Response (AR1)

**Response to reviewers**

**The TropD software package: Standardized methods for calculating Tropical Width Diagnostics**

Adam et al.

Executive comment by Astrid Kerkweg: Please add a version number for the software package TropD in the title upon your revised submission to GMD

A version identifier was added to the title (v1)

We thank the reviewers for their comments. A detailed response is provided below.

**Reviewer 1**

This work presents a framework in MATLAB programming language to calculate the tropical width using the eight most common methods and metrics for this purpose. The package of code called TropD includes not only the source files where all methods covered in the article were implemented, but also includes examples, data and documentation.

TropD is provided under an open source licenses, so it can be used and modified without restrictions. In this way, users can adapt it to their own studies, making a significant contribution to the scientific community.

The article clearly explains each method implemented in the code package TropD. Each important function is implemented in a separate file with multiple options for its execution and its interface is properly documented. An example code and validation data are included to evaluate the use of TropD.

No significant shortcomings were found in the implementation of this code package. The source code is clear and well organized. All variables are documented and the comments included in the code make this code easy to understand. Therefore, I have no suggestions for improving the code package presented in this work.

Thanks for the kind words.

**Reviewer 2**

This paper describes a software package that implements eight commonly-used methodologies for the calculation of metrics of the tropical width. A set of optional parameters is made available for some methodologies. The code is written in the widely used MATLAB programming language, and its purpose is to provide standardized calculation methodologies for commonly used diagnostics of the tropical width, helping in the comparison of the results from different studies.

The paper is well written and the software functions well described. A sensitivity analysis for the different methodologies is presented to justify the choice of the default options. Because the basic operators applied in the metric calculations are nonlinear, the metric calculations do not commute in space and in time. The authors discussed this in the manuscript. However, there is another effect of nonlinearity that the authors did not discuss. The input data for the software are zonal-mean variables in a latitude pressure grid. For example, in the TBP method, the zonal-mean tropopause is calculated by applying the WMO lapse rate criteria to the isobaric zonal mean temperature. I wonder if the results will be the same when the zonal mean tropopause parameters are obtained by zonally averaging the respective values calculated at each longitude.

Indeed, as the reviewer suggests, for some metrics, the metric derived from a zonal-mean field may differ from the zonal average of the metric applied per longitude. However, as

stated in Section 2.1, TropD is designed to be applied on zonal-mean fields. This is because the methodologies for zonally varying indices of the tropical width require more research, potentially followed by modifications of the code and methods. We may address this issue and incorporate methodologies for zonally-varying metrics of the tropical width in later versions of TropD.

I have more two minor comments:

pg. 3, line 16: "(ii) In cases where multiple zero-crossing latitudes exist, the first zero crossing along the input interval is chosen."

Because the data grid is ordered from the South Pole to the North Pole, the first zerocrossing latitude in the SH is far from the equator than the first zero-crossing latitude in the NH. Is it right?

A clarification note was added to section 2.2 (latitude of zero crossing). The comment reads: "The metric functions described below automatically order the input interval lat such that the first latitude of zero crossing in each hemisphere corresponds to the most equatorward zero crossing."

Fig. 9: It will be useful to call the attention for the different vertical scales used in Figs. 9a,b and 9c,d.

The caption of Fig. 9 was edited to note the differences between the vertical scales of the STJ:core and STJ:adjusted methods

**The TropD software package (v1): Standardized methods for calculating Tropical Width Diagnostics**

Ori Adam1, Kevin M. Grise2, Paul Staten3, Isla R. Simpson4, Sean M. Davis5,6, Nicholas A. Davis5,6, Darryn W. Waugh7, Thomas Birner8,9, and Alison Ming10

1Hebrew University of Jerusalem, Israel
2University of Virginia, USA
3Indiana University, USA
4National Center for Atmospheric Research, USA

[revised manuscript text omitted]

**Figure 3.** The dependence on n of the error distribution of  $\phi_{max}$  calculated using Eq. (1) in a representative sample of 100 randomized skewed Gaussian functions such as the one shown in Fig. 2. The error (gray dots) is defined as the difference between  $\phi_{max}$  and the latitude of the maximum of the smooth Gaussian function (black line in Fig. 2). Standard deviations of the error (STD(Error), horizontal lines) and histograms (normalized between 0 and 1) are shown for the error distributions of the sample for each n.

Figure 4. The mean tropopause height (a) and the difference in the potential temperature between the tropopause level and the surface (b) during the decades beginning in 1979 (green) and 1995 (orange) in CMIP5 models. The shading indicates  $\pm 1$  standard deviation of inter-model spread. The calculations are derived from monthly means of the temperature field.